# Drainless Thoracoscopic Lobectomy for Lung Cancer

**DOI:** 10.3390/jcm10163679

**Published:** 2021-08-19

**Authors:** Luo-Sheng Yong, Mong-Wei Lin, Ke-Cheng Chen, Pei-Ming Huang, Jang-Ming Lee

**Affiliations:** Department of Surgery, National Taiwan University Hospital, Taipei 100, Taiwan; kelvin_yls@hotmail.com (L.-S.Y.); ntuhmwl@gmail.com (M.-W.L.); cskchen@gmail.com (K.-C.C.); e370089@gmail.com (P.-M.H.)

**Keywords:** drainless, lobectomy, lung cancer, minimally invasive surgery

## Abstract

OBJECTIVES: Drainless video-assisted thoracoscopic (VATS) wedge resection has been demonstrated as feasible in treating various lung diseases. However, it remains unknown whether this surgical technique can be effectively applied to lobectomy. In the current study, we evaluated the perioperative outcome of drainless, minimally invasive lobectomy in patients with lung cancer. METHODS: A total of 26 lung cancer patients who received surgery-performed pulmonary lobectomy were enrolled. The perioperative outcomes were analyzed based on a propensity score matching a comparison with those who had chest drainage. RESULTS: No major surgical morbidity and mortality was noted during the perioperative period. The mean of postoperative hospital stay was 5.08 ± 2.48 days. There was no significant difference in postoperative hospital stay between the two groups of patients. However, the presence of significant postoperative pain (VAS score > 30) on the first day after surgery was less in the drainless group (34.6% vs. 3.8%; *p* = 0.005). CONCLUSIONS: Our results demonstrated that drainless, minimally invasive lobectomy for selected lung cancer patients is feasible. Further evaluation of its impact on short- and long-term surgical outcomes is required in the future.

## 1. Introduction

With the enormous development of video-assisted thoracoscopic surgery (VATS) in recent decades, a chest drain tube is not placed routinely anymore in selected patients and surgeries, especially in VATS pulmonary wedge resection, despite the fact that air leak remains the most common postoperative complication, with a reported rate of 5.6~40% [1,2,3]. Such a procedure of omitting chest-tube placement (drainless technique) can potentially reduce postoperative pain [4,5,6,7], preserve more ventilation capacity [5] and facilitate early ambulation [7]. However, this surgical technique has not yet been reported in a more complicated thoracic surgical procedure such as pulmonary lobectomy, which increases the residual air space after more extensive tissue dissection and for which placement of chest drainage tube was routinely used for prevention of subsequent pneumothorax or pleural effusion in treating lung cancer. For the first time in literature, our current study analyzed the perioperative outcomes in a series cohort and compared them with those done by the traditional minimally invasive thoracic procedure with drain tube placement.

## 2. Materials and Methods

The medical records of 116 patients who underwent pulmonary lobectomy performed by a single surgeon (J.M.L) at National Taiwan University Hospital between January 2017 and December 2019 were retrospectively reviewed, according to the STROBE statement [8], a checklist of which can be found in the Appendix A. Of these patients, 100 pathologies reported lung cancer, yielding mostly as adenocarcinoma (*n* = 89). Patients whose final pathology report showed diseases other than lung cancer were excluded (*n* = 16). The indications for pulmonary lobectomy were clinical T1–T3 disease, N0–N1, single station N2, and absence of distant metastasis as the guidelines suggested. Sex, age, smoking history, comorbidities, T and N stage of the tumor, the tumor location, and the surgical methods were recorded.

We initiated the practice of omitting chest tube placement in pulmonary lobectomy patients in November 2017. Chest tubes were not placed in a total of 26 patients (drainless group). The following criteria were used to select patients for drainless procedure: (1) No previous ipsilateral thoracic surgery. (2) No pulmonary inflammatory disease, as it could be associated with pleural adhesion during surgery. (3) No previous history of chronic obstructive pulmonary disease (COPD). (4) Preoperative FEV1 70% of prediction or more. The Research Ethics Committee of the hospital approved this retrospective study (202007047RIND) and waived informed consent.

### 2.1. Surgery

The thoracoscopic lobectomy was performed via single port approach [9], whereas the robotic-assisted lobectomy was performed under the Da Vinci Si or Xi system with three robotic arms and a 3-cm utility incision along the anterior costophrenic region as the assistant port and specimen retraction [10]. The wound would be enlarged if necessary during retraction of the specimen according to the size of the specimen. Lymph node dissection for the pulmonary hilum and mediastinum was performed for every patient after pulmonary lobectomy (Figure 1). A 28Fr. chest tube would be placed at the posterior aspect of the pleural cavity before wound closure.

### 2.2. Drainless Surgical Technique 

Among those patients who underwent the drainless approach, suturing was performed over the stapled raw surface of the remaining lobe with 4-0 polypropylene (PROLENE^®^|Ethicon). After ensuring that no air leaks were noted in the water-sealing test, a Jackson-Pratt (JP) drain was placed and the lung was re-ventilated under direct vision of the thoracoscope. The incision wound was then closed continuously, and the drain was connected with the JP ball to create negative intrapleural pressure. If the JP ball remained stably retracted in 2 min, the drain would be removed. 

### 2.3. Postoperative Management

Two postoperative chest X-rays were given for the patients who received the drainless surgical technique 5 h after surgery and the following day, respectively. In our institute, all patients who underwent pulmonary surgery received analgesic treatment of paracetamol 1 g 4 times daily and an NSAID (Celebrex 200 mg) twice daily. The pain experienced after surgery was evaluated with a visual analogue scale (VAS) presented as a 100 mm line aided by the description from “no pain” in the bottom to “worst imaginable pain” on the top [11]. Because postoperative with a VAS score less than 3 (30 mm/100 mm) was found to associated with the 75th percentile of good recovery after surgery [12] and more-active pain control is recommended for those with a VAS more than 3 (30 mm/100 mm) after surgery [13], we used VAS 3 as the cut-off value in comparing the different surgical groups in the propensity matching study. 

### 2.4. Statistical Analysis

The categorized demographic and clinical data among different surgical groups were compared using the c2 test or Fisher exact test. The mean values of continuous variables were analyzed by an independent sample *t*-test. A propensity score matching analysis was given for the comparison of the patients with or without drainage by matching the potential factors for influencing the perioperative surgical outcome.All statistical analyses were performed using statistical software package SPSS 17.0 (SPSS Inc., Chicago, IL, USA). 

## 3. Results

Drainless lobectomy was performed successfully in 26 lung cancer patients. No major surgical morbidity and mortality was noted during the perioperative period. The clinical parameters and results of patients undergoing lobectomy are demonstrated in Table 1. Patients’ baseline characteristics showed no significant difference between groups with and without chest-tube placement. Regarding the aspects of lung-tumor characteristics, no differences were noted in the tumor location, T stage and N stage on the final pathological results. Furthermore, the group with chest-tube placement had longer hospital stays (8.39  ±  10.23 days vs. 5.08 ± 2.48 days, *p* = 0.011) and higher frequency of a postoperative pain score >3, though the patients who underwent thoracotomy procedure were also included in this group (24.3% vs. 0.038%, *p* = 0.022).

### 3.1. Propensity Score Matching

Propensity score matching was performed and was used to create matched pairs comparable for age, gender, body mass index, T stage, N stage, tumor location, and surgical methods (Table 2). In fact, our study revealed that there is no difference in the length of postoperative hospitalization between the group with chest-drain placement and the group without. The patients without chest-drain placement had a lower rate of significant postoperative pain (VAS > 3) yielding a *p*-value of 0.005. (Figure 2).

### 3.2. Perioperative Complications

Subcutaneous emphysema and residual pulmonary atelectasis were detected in 57.59% and 23.18% of the patients with chest X-ray, which resolved spontaneously without any further intervention. Pleural effusion was detected in the postoperative day for 1 in 8 patients (30.7%) who received single thoracentesis for a single time in four cases (15.4%). No further chest-tube insertion was needed in our patients. There were also no in-hospital deaths or infectious events in these patients. Among all of the patients with drainage, subcutaneous emphysema and residual pulmonary atelectasis were noted in 29.73% and 21.62 of the patients, respectively, and residual pleural effusion was in 6.76% of the patients. Pneumonia developed in 2.7% of the patients. (Table 3).

## 4. Discussion

Adequate drainage is essential for general surgical procedures in preventing complications associated with surgical fluid accumulation. However, previous reports demonstrated safety in selected patients undergoing pulmonary wedge resection without pleural drainage tube placement [1,2,3,14,15]. In the current study, the drainless technique was applied in 26 lung cancer patients who underwent lobectomy. There was no major surgical complication related to this drainless procedure. Pleural effusion was detected in 30.7% (8/26) of the patients, among whom 15.4% (4/26) received a single episode of thoracentesis. However, no patient required chest drain re-insertion. In addition, the patients without chest-tube placement had a lower rate of significant postoperative day 1 pain (VAS > 3). This observation was in concordance with the results of other studies for pulmonary wedge resection or major lung resection [1,2,3,14,15]. It was found that patients with a VAS pain score more than 3 are more vulnerable to deterioration of life quality, and more aggressive pain control is recommended for these patients. Whether the reduction of pain in the early perioperative period can be translated into a reduction in postoperative complication and improvement in life quality requires further investigation. 

Atsushi Watanabe established the criteria for omitting chest-tube placement in pulmonary wedge resection [3]: (1) absence of air leaks during intraoperative alternative sealing test, (2) absence of bullous or emphysematous changes on inspection, (3) absence of severe pleural adhesions, and (4) absence of prolonged pleural effusion requiring chest drainage preoperatively [2]. Accumulating experience in the drainless technique for minimally invasive pulmonary wedge resection in our center has allowed us to extend the technique to a more extensive pulmonary surgical procedure such as pulmonary lobectomy. To prevent and reduce air leakage after lobectomy in patients with lung cancer, a precise preoperative evaluation and adequate intraoperative management are crucial for the success of drainless minimally invasive pulmonary lobectomy. The preoperative evaluation includes a lung-function test and transthoracic echocardiogram, and imaging by radiological examination helps to identify the high risk of patients for air leakage or massive pleural effusion after surgery and to modify the decisions for the surgical procedure, including the fine fissureless technique [16,17,18,19]. In agreement with the previous studies [20], an asymptomatic subcutaneous emphysema and residual pneumothorax are frequently detected on chest X-ray (Figure 1), none of which required a further drainage procedure after surgery. The tip of the intraoperative JP drain is suggested to be placed around the space around the intra-thoracic apex to ensure a complete lung expansion after surgery. There is no standardized method to detect intraoperative air leakage immediately after completion of the surgical procedures, including the use of a chest tube connected with a chest bottle [3,6,21], a chest tube connected with a suction system [14], a two-lumen catheter placement [22], and a digital chest drainage system [23,24]. From our experience, the negative pressure created by the J-P ball intraoperatively could provide a simple way to identify air leaks after surgical procedures of pulmonary lobectomy and help ensure the success of the drainless technique after surgery. 

Several strategies have been introduced in preventing air leaks after pulmonary lobectomy, including the use of tissue sealants or buttressing staple lines [24,25,26,27]. However, no sealants were applied after surgery in the current series. Although there were patients presenting with self-limited subcutaneous emphysema, no patients required further drainage intervention after surgery. It is our belief that a gentle and delicate tissue dissection during pulmonary lobectomy contributes to reducing possibilities of lung parenchyma injury and increases the chances of the drainless procedure, despite the absence of tissue sealants. The clinical value of using tissue sealant in these patients requires evaluation by evidence provided by a prospective randomized trial in the future. 

Our study is limited by its retrospective nature and the constraint from single-center series cases. The patients under careful selection for the drainless surgery do not represent the general population receiving pulmonary lobectomy. In particular, this study excludes the patients with COPD history or emphysematous change of lung on inspection during surgery, which might be more frequently present in the countries where smoking accounts for the major risk factor for developing lung cancer [28]. In addition, it is unclear about other specific surgical intervention procedures, i.e., the lymph node dissection or vascular control, etc., apart from the routine pulmonary lobectomy procedure, also has an impact on an uneventful drainless postoperative course. Therefore, the results should be carefully interpreted. 

In conclusion, the results of the current study indicate that drainless procedure is feasible for selected patients undergoing minimally invasive pulmonary lobectomy. This technique may reduce the presence of significant wound pain on the first postoperative day. Nevertheless, further studies are warranted to confirm the clinical value of drainless lobectomy in treating lung cancer.

## Figures and Tables

**Figure 1 jcm-10-03679-f001:**
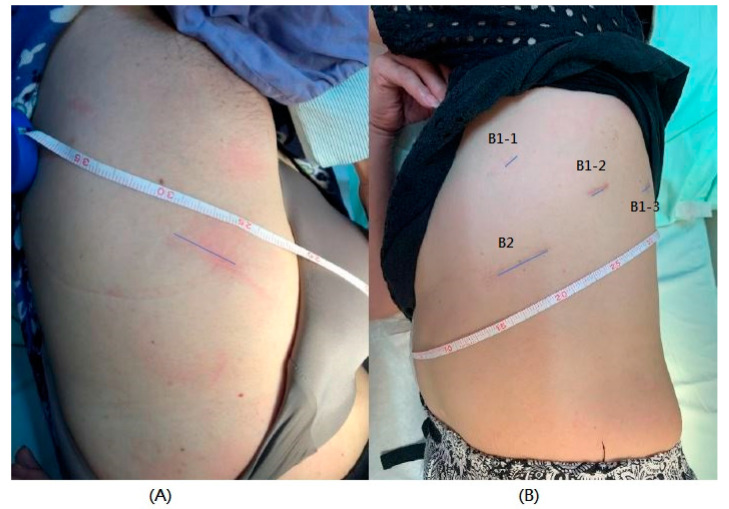
The port placement of VATS (**A**) and RATS (**B**). The thoracoscopic lobectomy was performed via single port (A) approach, whereas the robotic-assisted lobectomy was performed under the Da Vinci Si or Xi system with three robotic arms (B1-1, B1-2, B1-3) and a 3-cm utility incision (B2) along the anterior costo-phrenic region as the assistant port and specimen retraction. The wound would be enlarged if necessary during retraction of the specimen according to the size of the specimen ((**A**) and B2).

**Figure 2 jcm-10-03679-f002:**
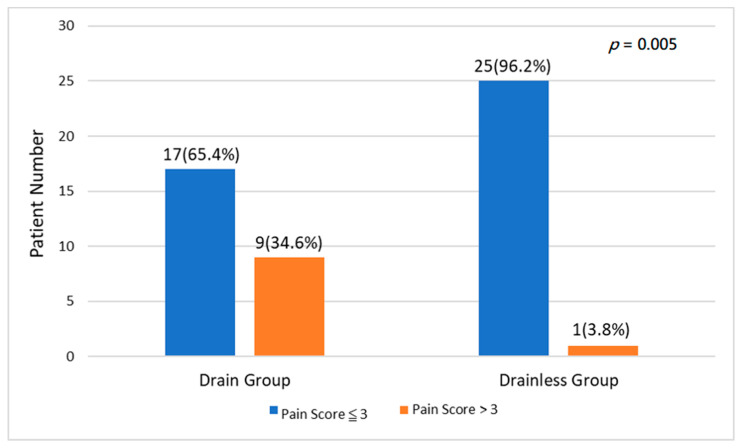
The comparison of pain score (<3 and >3) between the patients of minimally invasive lobectomy (VATS and RATS) with or without drainage after propensity score matching. (*p* = 0.005 for statistical difference).

**Table 1 jcm-10-03679-t001:** Characteristics of patients in the drain and drainless groups before propensity score matching.

Variable	Total	Drain	Drainless	*p*-Value
	*n* = 100	*n* = 74	*n* = 26	
**Sex**				0.090
Male	41	34	7	
Female	59	40	19	
**BMI**				0.934
<25	57	42	15	
≥25	43	32	11	
**HTN**				0.111
Yes	30	19	11	
No	70	55	15	
**COPD**				0.551
Yes	1	1	0	
No	99	73	26	
**Smoking history**				0.797
Yes	21	16	5	
No	79	58	21	
**T stage**				0.053
T3, T4	21	19	2	
T1, T2	79	55	24	
**N stage**				0.083
≥1	33	28	5	
0	67	46	21	
**Tumor location**				0.290
RUL	40	31	9	
RML	14	11	3	
RLL	9	4	5	
LUL	20	16	4	
LLL	17	12	5	
**Pain score (VAS)**				**0.022**
≤3	79	54	25	
>3	19	18	1	

**Table 2 jcm-10-03679-t002:** Characteristics of patients in the drain and drainless groups after propensity score matching.

Variable	Total	Drain	Drainless	*p*-Value
	*n* = 52	*n* = 26	*n* = 26	
**Sex**				0.760
Male	15	8	7	
Female	37	18	19	
**BMI**				0.578
<25	18	13	15	
≥25	24	13	11	
**HTN**				0.244
Yes	18	7	11	
No	34	19	15	
**COPD**				1
Yes	0	0	0	
No	52	26	26	
**Smoking history**				0.714
Yes	9	4	5	
No	43	22	21	
**T stage**				1
T3, T4	4	2	2	
T1, T2	48	24	24	
**N stage**				0.510
≥1	12	7	5	
0	40	19	21	
**Tumor location**				0.908
RUL	20	11	9	
RLL	5	2	3	
LUL	8	3	5	
LLL	9	5	4	
RML	10	5	5	
**Surgical approach**				1
VATS RATS	42 10	21 5	21 5	
Thoracotomy	0	0	0	
**Hospitalization**				1
≤7	44	22	22	
>7	8	4	4	

**Table 3 jcm-10-03679-t003:** Early postoperative complication among the drainless group.

Event	Incidence	Therapy Strategies
	No/Total (%)	
Residual pneumothorax	6/26 (23.18)	Observation and oxygen therapy
Pleural effusion	8/26 (30.77)	Observation (6) or thoracentesis (2)
Subcutaneous emphysema	15/26 (57.59)	Observation and oxygen therapy
Pneumonia/Empyema	0/26 (0)	

## Data Availability

Not applicable.

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
