# Peer review of "Drainless Thoracoscopic Lobectomy for Lung Cancer"

_jcm, 2021, doi:10.3390/jcm10163679_

Round 1

Reviewer 1 Report

I would like to congratulate the authors on the interesting paper titled “Drainless thoracoscopic lobectomy for lung cancer.”

Lung resection surgery without the use of a pleural drain has the potential to be very beneficial for patients, but may also lead to an increased risk of postoperative complications. Drainless thoracic surgery has not been discussed more widely in the scientific literature so far. For these reasons, the topic is very interesting and up-to-date.

The paper is written in clear and concise way, mostly with proper English. Some minor language editing could be useful.

In the “Introduction” section the authors cover the present state of knowledge on the topic. The authors should add one sentence regarding the safety concerns of the method, that is, its possible association with the higher incidence of residual air space and the need to re-drain patients. The aim of the study is clearly stated.

“Materials and methods” include almost all the elements required: ethics committee approval, type of the study, location, time frame, design, groups, inclusion criteria, data collection and statistical analysis. In the “Statistical analysis” section, a sentence about method of propensity score matching should be added.

Results are presented in proper way. Discussion is thorough, the authors analyze the findings presented in other papers and compare them to the results of their study. Limitations of the study are stated.

However, I believe that the authors should mention that a limitation of the study is also a small group of smokers and people with COPD. In the geographic region from which the manuscript comes, a significant proportion of lung cancer has a genetic basis. On the other hand, in many countries, smokers account for up to 95% of lung cancer patients, and the occurrence of COPD and emphysema is very common.  This means that the wide introduction of the method in countries with a high percentage of COPD and emphysema could be associated with an increased risk of complications in patients. Although the criteria for applying the method are known and were clearly described by the authors, I believe that for patient safety, one or two sentences concerning the limitation of using the drainless lobectomy in this group of patients should also be added to the limitations of the study.

The clinical significance of the study is high. I can recommend the manuscript for the publication in the Journal of Clinical Medicine after minor revision.

Reviewer 2 Report

 Thank you for the opportunity to review this interesting article.

 This study was investigated that the perioperative outcome of drainless lobectomy via single-port VATS or RATS in lung cancer patients. The authors showed that pain on the first postoperative day was significantly lower in the drainless group and concluded that drainless lobectomy for selected lung cancer patients is feasible. I agree with the conclusions of this study.

I have some comments and questions as follows:

MATERIAL AND METHODS section

  1. Surgery

 Please show the schema for both VATS and RATS approaches.

  1. Statistical analysis

 PSM has been performed in this study, but there is no description of it. Moreover, the number of cases in this study is small, so how significant is it to perform PSM? It needs to be confirmed by a statistician.

RESULTS section

  1. Table 1, 2

 It is better to provide a breakdown of VATS and RATS.

  1. Figure 1

 It is not specified which population is being compared. Before matching? Post-matching? Is RATS included?

  1. Perioperative complications

 How often did similar events occur in the drain group?

DISCUSSION section

 The discussion section was well written, but many of the statements are obvious and seem redundant. I think it should be a bit more concise and have fewer literature citations.

Round 2

Reviewer 2 Report

I think it has been properly fixed.

If possible, please create a figure for the port placement for VATS and RATS.

Author Response

Thanks so much for the comments of our revised manuscript. All of the comments of editors and reviewer has been addressed as follows.

  1. We had added the picture of ports-sites of VATS and RATS of the patients. Please have a check in the figure.